# Evaluation of Essential Oils and Extracts of Rose Geranium and Rose Petals as Natural Preservatives in Terms of Toxicity, Antimicrobial, and Antiviral Activity

**DOI:** 10.3390/pathogens10040494

**Published:** 2021-04-19

**Authors:** Chrysa Androutsopoulou, Spyridoula D. Christopoulou, Panagiotis Hahalis, Chrysoula Kotsalou, Fotini N. Lamari, Apostolos Vantarakis

**Affiliations:** 1Department of Public Health, Faculty of Medicine, University of Patras, 26504 Patras, Greece; chrysandr@gmail.com (C.A.); chrysoulakotsalou@gmail.com (C.K.); 2Laboratory of Pharmacognosy & Chemistry of Natural Products, Department of Pharmacy, University of Patras, 26504 Patras, Greece; spiridoulaxr@windowslive.com (S.D.C.); flam@upatras.gr (F.N.L.); 3Tentoura Castro-G.P. Hahalis Distillery, 26225 Patras, Greece; hahalis@tentoura.gr

**Keywords:** aromatic plants, essential oils, extracts, antimicrobial, antiviral, acute toxicity, lifetime

## Abstract

Essential oils (EOs) and extracts of rose geranium (*Pelargonium graveolens*) and petals of rose (*Rosa damascena*) have been fully characterized in terms of composition, safety, antimicrobial, and antiviral properties. They were analyzed against *Escherichia coli*, *Salmonella enterica* serovar Typhimurium, *Staphylococcus aureus*, *Aspergillus niger*, and Adenovirus 35. Their toxicity and life span were also determined. EO of *P. graveolens* (5%) did not retain any antibacterial activity (whereas at 100% it was greatly effective against *E. coli*), had antifungal activity against *A. niger*, and significant antiviral activity. Rose geranium extract (dilutions 25−90%) (*v*/*v*) had antifungal and antibacterial activity, especially against *E. coli*, and dose-dependent antiviral activity. Rose petals EO (5%) retains low inhibitory activity against *S. aureus* and *S.* Typhimurium growth (about 20−30%), antifungal activity, and antiviral activity for medium to low virus concentrations. Rose petals extract had significant antibacterial activity at dilutions of 25−90%, especially against *E. coli* and *S.* Typhimurium, antifungal, and the most potent antiviral activity. None of the EOs and extracts were toxic in dilutions of up to 5% and 90%, respectively. Finally, all materials had a life span of more than eight weeks. These results support the aspect that rose petals and rose geranium EOs, and extracts, have beneficial antimicrobial and antiviral properties and they can be used as natural preservatives.

## 1. Introduction

Medicinal Aromatic Plants (M.A.P.) are a major part of the natural flora and considered an important resource in various fields, for instance the pharmaceutical, cosmetic, food, fragrance, and perfumery industries [1]. Nowadays more than 80% of the world population relies on traditional herbal medicines to treat health issues [2,3]. More than 9000 plants have been identified and recorded for their therapeutic properties. About 1500 species are known for their aroma and taste [4]. Natural Aromatic Chemicals are in great demand in different fields such as cosmetics, food, pharmaceuticals, and perfumes. On the contrary, today’s society is facing a negative opinion with the chemical preservatives. Therefore, organic foods are commonly produced without the use of chemicals, and their microbial load could be higher. Therefore, plant chemical compounds constitute an alternative to their maintenance [5,6]. EOs and extracts have been used for their aroma, taste, medicinal properties, and as bactericides and preservatives [7]. Therefore, EOs and extracts are a safe, environmentally friendly, cost-effective choice for nutrition and environmental protection. EOs and extracts have gained their role as preservatives due to certain chemical compounds they contain, such as terpenes, terpenoids, carotenoids, and phenolics [8,9]. Many EOs and extracts have anti-inflammatory, antibacterial, antifungal, antiviral, and antiseptic features [10,11,12].

*Pelargonium graveolens* (Thunb.) L’Hèr belongs to the Geraniaceae family and to the genus of *Pelargonium*, which includes more than 200 species. It is a perennial herbaceous species and has its roots in South Africa; however it is now growing in other places as an ornamental plant and is also cultivated for its use in the food and drink industry [13]. The main volatile compounds are oxygenated monoterpenes (64.3−74.2%), but the essential oil composition greatly depends on the genotype, the environmental conditions, the agricultural practices of cultivation, and the time of collection [14].

*Rosa damascena* Herrm. belongs to the Rosaceae family and the genus of *Rosa* [15]. The major constituents in the ΕO are β-citronellol, geraniol, eugenol, 2-phenylethyl alcohol, methyleugenol, linalool, and aliphatic hydrocarbons [16]. The antimicrobial, antioxidant, analgesic, anti-inflammatory, antidiabetic, and antidepressant properties of *R. damascena* have been demonstrated in preclinical studies [17]. Its petals have attractive organoleptic properties and find numerous applications in cooking. They are also included in numerous food products and drinks.

The Food and Drug Administration (FDA) and the Environment Protection Agency (EPA) in the USA have recognized many essential oils as “safe products” for food and beverage consumption [18]. However, their adjustment as food preservatives demands detailed awareness about their properties and concentrations.

The purpose of this study was to fully characterize the chemical composition of EOs and extracts from rose petals and rose geranium, and test their efficacy against *E. coli*, *S. aureus*, S. typhimurium, *A. niger*, and Adenovirus 35 in different concentrations. Their toxicity levels were determined, to be used as food preservatives without causing toxicity problems in the human body. Finally, their life span was determined to establish implementation of EO and extract constituents as natural food preservatives.

## 2. Materials and Methods

### 2.1. Essential Oils and Extracts

Petals of *Rosa x damascena* Herrm. (rose petals) originated from Agios Georgios Sikousis, Chios island (Northern Aegean, Greece) and were collected in May–June 2019, while *Pelargonium graveolens* (Thunb.) L’Hèr (rose geranium) was collected from a cultivation from the area of Patras (Achaia, Peloponnese, Greece) in May–June 2019.

The EOs were isolated with water steam distillation in an experimental 10 L distillery. Specifically, 0.6 kg of geranium leaves and 0.56 kg of rose petals were distilled in a final volume of 8 L for 3−4 h.

The herbal extracts were produced by maceration. The extraction took place in glass and stainless-steel containers. Fresh rose petals were dried for 1 day and 2 kg of plant material were extracted for 35 days, at a temperature of 22−27 °C, in 40 L of water containing 15% *v*/*v* ethanol. Fresh geranium leaves were dried for 8 h and then 0.2 kg of plant material were extracted for 30 days, at a temperature of 22−27 °C, in 40 L of 38% *v*/*v* aqueous ethanol.

### 2.2. Gas Chromatography-Mass Spectrometry

All samples were analyzed using an Agilent Technologies 6890N equipped with a 5975B mass selective detector (MSD) in the electron impact (EI) mode of 70 eV. The capillary column was HP-5MS (30 m × 0.25 mm, 0.25 μm) with helium as a carrier gas. Data analysis was performed with GC/MSD Chemstation (Agilent Technologies Inc., Santa Clara, CA, USA) and Mestre Nova v.6.0.2-5475 (Mestrelab Research S.L., Santiago de Compostela, Spain).

The analysis of *P. graveolens* leaf EO was performed according to Sharopov et al. [19], with small modifications. In brief, the initial GC oven temperature was 56 °C for 2 min and then ramped at 3 °C /min to 200 °C, and finally at 2 °C/min to 220 °C for 3 min. Carrier gas was at a rate of 1.0 mL/min in a splitless mode and the *m*/*z* range was 35−400.

For the *R. damascena* petals’ EO analysis, the initial GC oven temperature was 50 °C for 3 min, which was then ramped at 5 °C /min to 100 °C for 3 min, increased to 150 °C at a rate of 3 °C /min and was held constant for 1 min. Finally, the oven reached the temperature of 280 °C at a rate of 12 °C /min and then was kept at 280 °C for 3 min. Carrier gas was at a rate of 1.0 mL/min, in a splitless mode and the *m*/*z* range was 40−1000.

All samples were diluted (1:40 for rose petals EO and 1:30 for geranium leaf EO) in ethyl acetate and the injection volume was 1 μL. n-Octane (98% purity) was used as an internal standard (final concentration 0.3 mg/mL). Alkanes (C8−C24) were analyzed under the same conditions and were used as reference points for the calculation of retention indices with the Van den Dool and Kratz equation [20]. Identification of the chemical components was based on comparison of the experimental retention indices (AIexp) and the obtained MS spectra to commercial databases [21,22] and the literature. Results were expressed as the percentage of the ratio of each compound peak area to that of the internal standard, using the program WSEARCH32 (Ver. 16/2005). Only compounds with peak area higher than 0.1% are presented.

### 2.3. Liquid Chromatography-Mass Spectrometry

The samples for UPLC-ESI-MS were prepared by dilution from the original concentrations of the extracts to a final volume of 500 μL. In detail, the initial geranium leaves’ extract, 5.00 mg/mL, was diluted to a final concentration of 4.75 mg/mL, and the rose petals’ extract was diluted to a final concentration of 25 mg/mL from the original 50 mg/mL. Both samples included 50 μL 1% formic acid in a final volume of 500 μL.

The single quadrupole LC/MS system of LC/MSD1260 Infinity II (Agilent Technologies Inc., Santa Clara, CA, USA) was used in this study. This system was equipped with an ESI ion source and the mass range was *m*/*z* 100−1000. Nitrogen was applied as gas for ionization. Separation was performed on a Poroshell 120 EC 18 column (4.6 × 100 mm, 2.7 μm) (Agilent Technologies Inc.). LC conditions were as follows: solvent A (0.1% formic acid) and solvent B (acetonitrile with 0.1% formic acid). A gradient elution was used as follows: 0−5 min 4% B; 5−15 min 4−15% B; 15−18 min 15% B; 18−23 min 15−20% B; 23−33 min 20% B; 33−48 min 20−58% B; 48−63 min 58% B; 63−75 min 58−95% B; 75−80 min 95% B; 80−84 min 95−4% B; 84−88 min 4% B. Flow rate was 0.5 mL/min and the injection volume was 20 μL.

The standards that were used for identification were rutin (HPLC > 99%) from Extrasynthese (Genay, France), quercetin 3-O-glucoside (HPLC > 98%) from Extrasynthese (Genay, Gedex), and kaempferol (HPLC > 96%) from Sigma Aldrich (Steinheim, Germany). The identification of the other compounds was based on comparison of their retention time and their mass spectra to the literature. The quantification analysis of the compounds was based on the rutin standard curve (3.125−100.000 μg/mL, y = 29361x + 425743, R^2^ = 0.9632). The Lower Limit of Quantitation (LLOQ) was 3.125 μg/mL since it is the lowest concentration of rutin giving an acceptable accuracy (relative error <20%) and the Lower Limit of Detection (LLOD) was 0.875 μg/mL calculated as a signal to noise ratio of 3.

### 2.4. Mutagenicity Assay

The toxicity of the samples was determined by Ames Salmonella/mutagenicity assay (Salmonella test, Ames test), (EBPI, AirMetal, Los Angeles, CA, USA). The Ames test is a short-term bacterial reverse mutation trial especially designed to track a variance of chemicals that can cause genetic damage that conducts to gene mutations [23,24].

The Salmonella tester strains were TA98 and TA100. TA98 causes frameshifts [25], while TA100 provokes base-pair substitution [26]. The lyophilized strains (2.5 μL) were transferred to Growth Media containing 5 mL, 0.01 mL of Express Reagent ‘V’, (different vial for each strain). Then, they were incubated at 37 °C for 18 h. The next day, the bacterial growth was checked for turbidity. The experimental procedure was continued only if there was turbidity [27].

Τhe test was performed by the pre-incubation method [23]. From each colony cultivated overnight, 0.1 mL is taken and mixed with 0.1 mL of the potential mutant at various concentrations. For the EOs, the concentrations were up to 5%, while for the extracts up to 90%, since these are the usual upper limits of their presence in foods and drinks. Then, 2 mL Top Agar with 10% sterile 0.5 mM L-histidine HCl, 0.5 mM biotin solution were added. The contents of each tube were poured into a sterile multi-channel reagent container and 200 µL of dilutions were distributed to each of the 24-well plates, using a multi-channel pipette. After incubation at 37 °C for 48 h, toxicity was determined. The samples were placed in closed bags, as it is proposed for testing highly volatile chemicals and gases [25,28,29,30]. *S.* typhimurium cannot grow in the absence of histidine [31,32]. If developed, it means that the substance is mutagenic. The EOs are dissolved in DMSO [33,34], while the extract samples are dissolved in distilled water. The determination of toxicity was colorimetric at 600 nm. A color change from purple to yellow signifies mutagenicity. All experiments were performed in triplicate at each concentration.

### 2.5. Antibacterial Assay

For the antibacterial activity, *E. coli* NCTC 9001, S. Typhimurium NCTC 12023, and *S. aureus* NCTC 6571 (SIGMA-ALDRICH) were grown on Brain Heart Infusion Broth (BHI Broth), (OXOID), at 37 °C for 24 h. As for the antifungal activity, *A. niger* (SIGMA-ALDRICH) was grown on Potato Dextrose Agar (PDA), (BIOlab), at 22 °C for 5 days [35].

The assay was performed by the agar dilution technique against a group of bacterial strains, as recommended by the National Committee for Clinical Laboratory Standards [36].

After placing 5 μL of different dilutions of the EOs (1, ½, ¼) and the extracts (90%, 50%, 25%) on Brain Heart Infusion Agar (BHI) plates, plates were dried at room temperature. Then, 1 μL of the bacterial strain (10^8^ CFU/mL), in serial dilutions, was spread to the plates. Plates were left to dry at room temperature for 30 min, before the incubation. The incubation was implemented at 37 °C for 24 h. After incubation, there was a record of the presence or absence of microorganisms and a comparison with Control. All experiments were performed in triplicate at each concentration.

### 2.6. Antifungal Assay

After *A. niger* was developed on Potato Dextrose Agar (PDA), the fungal cultures were replaced on PDA that contained 5 μL of different dilutions of the EOs (1, ½, ¼ dilutions) and the extracts (1/1.1, ½, ¼ dilutions). The plates were incubated at 22 °C for 7 days. Then, a daily monitoring of the diameter of each colony occurred and the results were recorded and compared with the Control. All experiments were performed in triplicate at each concentration. In the evaluation of the antifungal activity, the percentage inhibition was calculated.

### 2.7. Antiviral Assay

A549 cell culture (Life Science Chemilab, Athens, Greece), P +92 generation, were grown in 25 cm^2^ flask containing 40−50 mL Dulbecco’s Minimum Essential Medium (DMEM) containing 10% Fetal bovine serum (FBS) and 1% antibiotic antimycotic solution. Then, cells were transferred in 24-well cell culture microplates. Fetal bovine serum was used for the growth of cells [37]. The antibiotic was added in order to avoid contamination. The cell cultures were maintained in a humidified atmosphere, 5% CO_2_ at 37 °C.

Stock human Adenovirus serotype 35 was propagated in A549 cells. For titration of viruses, A549 cells were seeded in 24-well culture microplates and then incubated. Serial dilutions of virus were prepared in culture mediums. Each dilution was added into four of the wells. After incubation the cytopathic effect in each well was recorded.

First of all, the cytopathic effect of the samples was performed with the same process as the cytopathic effect of Adenovirus. Then, the analysis of the antiviral activity of EOs and extracts took place. Cell line A549 was incubated overnight in 12-well plates at 37 °C with 5% CO_2,_ until cells got “confluent 90−100%” overlap. The volume of each cell line on each well plate was calculated by counting the number of cells at the four corners of the Neubauer.

Then solutions containing serial concentrations of AdV (30 µL) and the sample (30 µL) were added and incubated with stirring (150 rpm) for 90 min at 37 °C. The medium was discarded and DMEM was re-added with 1% FBS. The 12-well plates were incubated for 3−4 days at 37 °C with 5% CO_2_. Subsequently, the cytotoxicity was observed by electron microscope. Tests were performed according to Saderi et al., 2011 [28], claiming that the concentrations of the samples that entirely suspend AdV35 cytopathic effect, is recorded as efficient concentration, comparing to virus control. All experiments were performed in duplicate at each concentration.

### 2.8. Life Span Assay

Τo determine the life span of the samples, samples were added to the proper medium. Samples were added to the Tryptone Bile X-Glucuronide Medium (TBX), to detect *E. coli*. Also, samples were added to the Rappaport Vassiliadis broth (RVS), to detect *Salmonella* spp. Furthermore, samples were added to the Baird-Parker Agar (BP), (OXOID), in order to detect *S. aureus*, and finally, were added to the Potato Dextrose Agar (PDA), (BIOlab), to detect *A. niger*.

In order to determine the time when the consumption of the products is safe, a three-week procedure was performed. The samples were kept in appropriate conditions, in the refrigerator at 10 °C for a period of five weeks, before the experimental procedure began. The experimental procedure was carried out for a period of three weeks, when the samples were stored in a refrigerator at 10 °C. All experiments were performed in duplicate at each concentration.

## 3. Results

### 3.1. Chemical Analysis

The chemical analysis of the metabolites of EOs and extracts is presented in the Table 1, Table 2 and Table 3. The evaluation of the mass spectra of each chromatographic peak did not allude to the presence of any pesticide or other contaminants in the samples.

### 3.2. Antimutagenesis

Antimutagenic effect regularly depends on the dose of the sample [38]. In this study, the doses of 5% for EOs and 90% for extracts were not toxic for the strains (Table 4). All EOs were able to inhibit mutations induced by TA98 and TA100 Salmonella strains, as well as all extracts at concentrations up to 90%.

### 3.3. Antibacterial Activity

The antibacterial activities of the different kinds of EOs and extracts were assessed by the agar dilution method. It can be seen from the data shown in Table 5, that all essential oils and extracts had some antibacterial activity on the tested strains, nevertheless the antibacterial properties varied significantly.

*P. graveolens’* essential oil showed a significant antibacterial activity (85%) against *E. coli* in 100% concentration. In lower concentrations, it did not affect *E. coli* growth. On the other hand, the extract showed good antibacterial activity (68%) against *E. coli* in 90% concentration and average efficacy in 50% and 25% concentrations (59% and 46% respectively). As for *S. aureus*, the essential oil had good antibacterial activity (55%) in 100% concentration. In lower concentrations, it was not effective. The extract was not effective against this bacterium—only in 90% concentration it appeared to have a minor inhibition (40%). Finally, the essential oil had a small activity against *Salmonella* spp. (51%), in 100% concentration. In lower concentrations, its activity was negligible. The extract was almost efficient in 90%, 42%, while in lower concentrations, it had no effect.

*R. damascena*’s essential oil showed a noteworthy antibacterial activity against *E. coli* (65%, in 100%). The extract revealed a remarkable antibacterial activity (74%, in 90% and 57% in 50%). Furthermore, the antibacterial activity against *S. aureus*, was only noticeable (68%) in the extract in concentration 90%. Last of all, the antimicrobial potency against *Salmonella* spp., showed drastic essential oil action (86% in 100% concentration and 73% in 50%), but also capable extract activity (86% in 90% concentration, 65% in 50% and 45% in 25%).

### 3.4. Antifungal Activity

Antifungal activity is presented in Figure 1a–c for EO and Figure 2a–c for extracts. The Figures below Figure 1a–c show the increase (in cm) in fungal diameter over the course of one week (7 days) in the presence of the essential oils at different percentages.

Experimental data show that all EOs in the concentrations of 100% and 50% were particularly effective against the fungus. At the end of the third day, the fungus stabilizes almost in diameter in all the oils and stops growing, while in the control it grows until it occupies the whole petri dish, which is why in the last days of the experiment its growth rate is low. As the days go by, the growth rate reaches saturation. The reason this happens is because natural preservatives contain nutrients that initially favor the growth of the fungus. Nevertheless, this particular resistance can disappear [40]. The EO of rose geranium has the greatest effect on the fungus. At lower concentrations rose geranium EO, even at a concentration of 5%, maintained activity. Figure 2 presents the increase (in cm) in fungal diameter over the course of one week (7 days) for *P. graveolens*’ and *R. damascena*’ extracts.

From the above experimental data, it appears that the 90% extracts were efficient. In this case also, the fungi grow until the third day and then their growth stops, while in the control sample the growth is limited by the petri dish and stops. Potato Dextrose Agar (PDA) provides a nutrient base for luxuriant growth of most fungi [41]. In 50% plant extracts, they have a moderate effect whereas in the 25%, plant extracts have similar effect.

### 3.5. Antiviral Activity

The results of cytotoxicity assay for tested samples are shown in Table 6. The cytopathic effects for EOs were observed in concentrations up to 5%, while the extracts did not show cytopathic effect even at 100% concentration.

Table 7 shows the results of experiments on EOs (5%) and extracts, against Adenovirus at concentrations of 10^9^ PFU/mL to 10^4^ PFU/mL.

### 3.6. Life Span

Concerning *E. coli*, throughout the experiment, no blue/gray colonies were detected on Tryptone Bile Glucuronic Agar (TBX agar), therefore all the samples are characterized as safe for human consumption, with regard to the *E. coli* microorganism. As for *Salmonella* spp., the experiments carried out showed that: at the weight of 25 g of the sample and 225 mL of Peptone Buffered Water, the microorganism was not detected, so the samples were considered safe for consumption, with respect to the *Salmonella* spp. Also, in all samples, *S. aureus* could not be detected. Finally, for *A. niger*, throughout the experiments, the samples are found to be free of mold and fungi and are considered safe for consumption.

## 4. Discussion

In the essential oil of geranium leaves, thirty-nine compounds were identified, which accounted for 67.98 ± 5.15% of the total oil. The main components of the essential oil were citronellol (26.7%) and geraniol (10.1%). The other main ingredients were α–pinene, *cis*–linalool oxide, *cis*–rose oxide, *trans*–rose oxide, menthone, isomenthone, α–copaene, β–bourbonene, and citronellyl propanoate; all those have been reported previously for geranium leaves’ EO [19].

A variability in the percentages of the main ingredients has been recorded. In a survey conducted in Greece and especially in the island of Crete, the two main components, citronellol and geraniol, were determined at percentages of over 25% and about 20%, respectively [42]. In another study of native geranium leaf oil from Tajikistan, citronellol and geraniol were present in percentages of 37.5% and 6%, respectively [19]. Accordingly, Verma et al. showed that the composition of the essential oil depends on the duration of cultivation; in that study, geraniol content was at its maximum in the crop transplanted in the month of April and citronellol content was higher in that transplanted in February, whereas all plants were harvested in June [43]. Moreover, differences of the ratio of citronellol and geraniol, according to the season of collection, have been reported; the ratio is close to one in April and June [42]. Finally, Wahab et al. [44] showed that the citronellol and the geraniol content was affected by planting location and harvest time. That work was carried out during the two successive seasons (2012/2013 and 2013/2014) to investigate the effect of different planting locations (5 different locations in Egypt) on quantity and quality of *P. graveolens* volatile oil. In most locations, the highest citronellol content was obtained in spring cut, while the lowest was obtained in autumn cut [44].

In the essential oil of rose petals, thirteen compounds were identified, representing 96.47 ± 0.43% of the total oil (Table 1). In our study, the main components of the essential oil were identified as phenyl ethyl alcohol (40.2%) and geraniol (27.8%). The remaining detected compounds were less than 20%. In accordance with our results, Verma et al. showed that phenyl ethyl alcohol was the main component in *R. damascena* EO of Ranisahiba cultivar (76%), and of Noorjahan and Kannouj cultivars (80.7 and 76.7%, respectively) at full bloom stage [45]. Similarly, in the GC analysis of EOs of *R. damascena* and *R. moschata* var. nastarana flowers by headspace extraction, phenyl ethyl ethanol was the dominant ingredient albeit in different amounts [46]. Interestingly, in the study of Koksal et al. [47] on damask roses from Turkey, phenyl ethyl alcohol was the major component in EO from fresh rose petals (25.06%), and that amount significantly increased when the plant material was stored before distillation [47]. The relationship of the chemical composition with the climate has also been suggested; according to Misra et al. [48], geraniol content decreases whereas phenyl ethyl alcohol increases in colder climates and at higher altitude [48]. Accordingly, in our study, rose petals were harvested in late spring from a mountainous location (360 m altitude) at a Northern Aegean island, which enjoys mild Mediterranean climate; in late spring the mean temperature does not exceed 20 °C. Therefore, differences in the rose petal EO composition can be attributed to several factors like the genotype, the cultivation, and the storage conditions.

The analysis of rose petals’ extract showed eighteen identified components. The main component of the herbal extract was kaempferol-3-O-glucoside (55.4%). The remaining detected components were less than 12%. Compounds were mainly identified as quercetin glycosides and kaempferol derivatives by comparison of their retention time with the standards and the literature data [49,50,51,52,53]. As can be seen from Table 2, the kaempferol glycosides accounted for the largest percentage of the compounds that were quantified, with kaempferol-3-O-glucoside being the predominant component. This is in accordance with other studies that report that kaempferol-3-O-glucoside is the main component in rose petals of *R. damascena* [51], and in rose petals of Taif rose [53]. In this study, nine kaempferol derivatives were identified. Most of them have also been reported in other species of the Rosacae family [49,50,52,53]. Kaempferol deoxyhexoside and kaempferol acetyldisaccharide were previously identified in extracts of *R. damascena* from Bulgaria [51]. Compounds 1 and 2 have also been determined from *R. damascena* from Egypt [54], and the hybrid “Jardin de Granville” from France [50], respectively. Five quercetin derivatives were identified in this study. Compounds 4 and 8 were also detected in *R. damascena* from Bulgaria [52] and from India [48]. The presence of quercetin glycosides and kaempferol aglycone has been reported also in other species of the Rosacae family, in addition to *R. damascena*, such as *Rosa bourboniana* Desport., *Rosa brunonii* Lindl., known as ‘Himalayan musk rose [49], the hybrid “Jardin de Granville” [50], and Taif rose, Ward Taifi (*R. damascena trigintipetala* Dieck) [53].

The geranium leaves’ extract analysis (Table 3) showed eleven identified components. The main components of the herbal extract were quercetin 3-O-glucoside (31.7%), quercetin-3-O-pentosyl hexoside (28%), and quercetin-3-O-galactoside (24.9%). The remaining detected compounds were less than 10%.

Most of the compounds were flavonol glycosides, i.e., quercetin and kaempferol derivatives. Compounds 1–9 were previously reported in a study on *P. graveolens* [55]. Compounds 2, 6, and 7 have been reported in other species of the Geraniacae family, except from geranium, i.e., *Geranium molle* L. [56] and *Geranium robertianum* L. [57]. Finally, to the best of our knowledge compounds 10, 11 have not been reported in Geraniacae family. Our results show that the quercetin glycosides accounted for the largest percentage of the compounds that were quantified, with quercetin-3-O-glucoside being the main component.

Our results are in line with previous studies demonstrating that rose oil and its aqueous extracts have moderate broad-spectrum antimicrobial activity [58,59]. Phenyl ethyl alcohol, the main rose oil constituent (40%) in our study, has been attributed antimicrobial properties for a long time but their inhibitory actions are considered complex and generally seem to be dominated by their physicochemical properties [60]. In addition, geraniol, the other major rose oil constituent (about 28%), and rose geranium EO constituent (10%), and is demonstrated to have antimicrobial activity against 78 different microorganisms, such as *Candida* or *Staphylococcus* [61]. A recent study by Guimaraes et al. (2019) [62], showed that geraniol and citronellol (the major rose geranium EO constituent and a major rose oil one) are fast-acting compounds that inactivate *E. coli* and *S.* Typhimurium by inducing loss of cellular membrane integrity or function. Thus, the antimicrobial activity of rose geranium oil demonstrated is our study is explained by, and is in agreement to, previous reports [63,64]. Concerning their antiviral activities, limited information is available about rose geranium oil, and no information is available, as far as we know, for citronellol, geraniol, and phenyl ethyl alcohol [65].

The major component of the extract of rose petals is kaempferol-3-O-glucoside, alongside with other kaempferol glycosides, and it has been shown to have antimicrobial and antioxidant effects [66,67,68]. The rose geranium extract, also rich in flavonoids (quercetin glycosides), had strong antimicrobial and antiviral activity in our study; our results are in agreement with earlier reports on the antimicrobial activity of geranium extracts [63,69] and quercetin derivatives [66]. The antiviral properties of quercetin and kaempferol derivatives have been reported in numerous publications [70], e.g., against influenza viruses [71,72], coronaviruses, and dengue viruses [73,74], and in particular of quercetin against adenoviruses 1 and 3 [75]; the reports indicate that those flavonols block viral entry to the host cell via specific interactions with viral attachment factors and/or membrane fusion proteins, suppress signaling pathways that are essential for virus gene expression, inhibit remodeling enzymes and channels (which regulate viral movement (e.g., neuraminidase)), and inhibit transcription of the viral genome and viral protein synthesis [70]. However, none of the previous studies have addressed the issue of their realistic application in foods and drinks as natural preservatives, since a final product will contain these EOs and extracts in a certain percentage, due to the organoleptic properties they confer, e.g., the strong odour.

## Figures and Tables

**Figure 1 pathogens-10-00494-f001:**
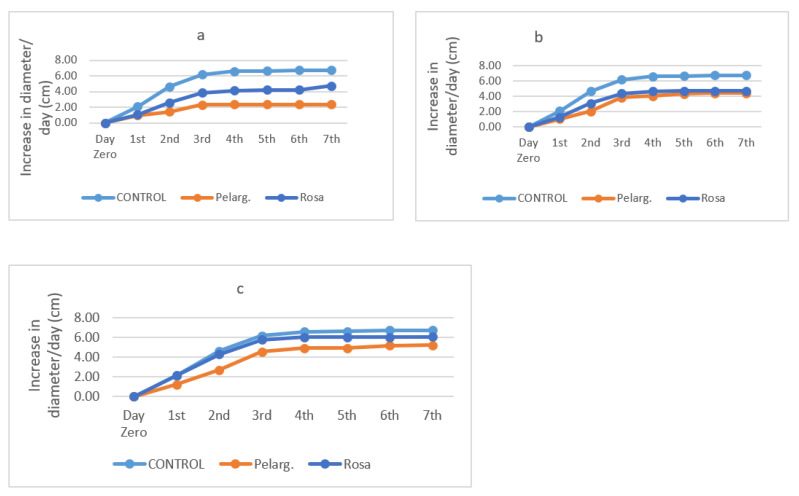
Antifungal activity of essential oils 100% (**a**), 50% (**b**), 5% (**c**) through time.

**Figure 2 pathogens-10-00494-f002:**
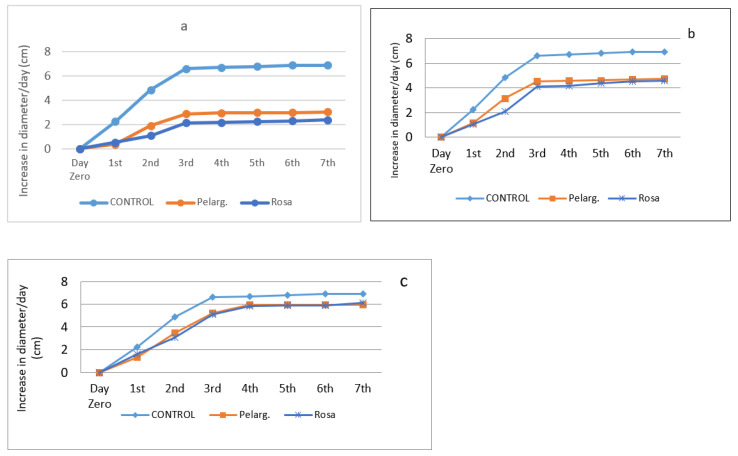
Antifungal activity through time of extracts at concentrations of 90% (**a**), 50% (**b**), and 25% (**c**).

**Table 1 pathogens-10-00494-t001:** Volatile metabolites in essential oils of the geranium leaves and rose petals.

Components	AIexp ^a^	AΙtheor ^b^	% Peak Area/IS Area	
			*Pelargonium graveolens*	*Rosa damascena*
α–pinene	929	932	0.19 ± 0.02	nd
*cis*–linalool oxide (fr)	1069	1067	0.21 ± 0.08	nd
*trans*–linalool oxide (fr)	1085	1084	0.17 ± 0.03	nd
linalool	1092	1095	2.93 ± 0.17	3.32 ± 0.17
*cis*–rose oxide	1108	1106	0.84 ± 0.06	nd
phenyl ethyl alcohol	1117	1107	nd	40.03 ± 0.13
*trans*–rose oxide	1125	1122	0.29 ± 0.01	nd
menthone	1150	1148	0.41 ± 0.06	nd
isomenthone	1163	1158	3.02 ± 0.09	0.41 ± 0.06
*α*–terpineol	1195	1186	0.16 ± 0.00	0.74 ± 0.01
methyl chavicol (estragole)	1205	1195	nd	0.59 ± 0.00
citronellol	1228	1223	26.73 ± 1.65	19.89 ± 0.34
neral	1239	1235	0.17 ± 0.00	nd
geraniol	1254	1249	10.1 ± 0.17	27.8 ± 0.90
citronellyl formate	1274	1271	3.96 ± 0.01	0.76 ± 0.27
geranyl formate	1300	1298	1.05 ± 0.06	nd
citronellyl acetate	1352	1350	0.14 ± 0.00	nd
α–copaene	1371	1374	0.33 ± 0.07	nd
β–bourbonene	1380	1387	0.81 ± 0.03	nd
geranyl acetate	1383	1379	0.16 ± 0.02	nd
β–elemene	1393	1389	nd	1.10 ± 0.00
vanillin	1399	1393	7.06 ± 1.79	nd
Ε–caryophyllene	1414	1417	1.12 ± 1.25	0.82 ± 0.00
aromadendrene	1439	1439	0.31 ± 0.07	nd
citronellyl propanoate	1442	1444	0.22 ± 0.01	nd
*allo*–aromadendrene	1456	1458	nq	nd
geranyl propanoate	1473	1476	0.46 ± 0.05	nd
γ–muurolene	1477	1480	nd	0.28 ± 0.05
E-*β*-ionone	1489	1487	nd	0.50 ± 0.26
α–muurolene	1501	1500	0.14 ± 0.00	0.28 ± 0.04
γ–cadinene	1509	1513	0.16 ± 0.00	nd
δ–cadinene	1519	1522	0.74 ± 0.48	nd
citronellyl butanoate	1526	1530	0.39 ± 0.01	nd
α–agarofuran	1540	1548	0.30 ± 0.14	nd
geranyl butanoate	1559	1562	0.49 ± 0.05	nd
spathulenol	1573	1577	nq	nd
caryophyllene oxide	1578	1582	0.24 ± 0.00	nd
phenyl ethyl tiglate	1584	1584	0.34 ± 0.03	nd
10–*epi*–*γ*–eudesmol	1614	1622	3.28 ± 0.38	nd
γ–eudesmol	1632	1630	0.25 ± 0.00	nd
4*a*-hydroxy-dihydro agarofuran	1643	1651	0.28 ± 0.00	nd
geranyl tiglate	1700	1696	0.74 ± 0.15	nd
hexadecanoic acid	1961	1959	0.65 ± 0.00	nd
number of components			39	13
total identified			67.98 ± 5.15	96.47 ± 0.43

Notes: IS: internal standard, nd: not detected, nq: not quantified. Results are presented as mean ± standard deviation derived from triplicate analysis. ^a^ Retention index on an apolar HP-5MS column. ^b^ Literature retention indices on apolar column from Adams et al., 2012 [21].

**Table 2 pathogens-10-00494-t002:** LC/MS identification of metabolites and their concentration in the extract of rose petals.

Peak	Rt (min)	Negative Ionization (*m/z*)	Positive Ionization (*m/z*)	M.W.	Molecular Formula	Tentative Identification	C (μg/mL)
1	1.8	179[M - H]^−^ 215[M + Cl]^−^ 217[M + K - 2H]^−^	203[M + Na]^+^ 383[2M + Na]^+^	180	C_6_H_12_O_6_	Hexose ^52^	5.3 ± 0.4
2	1.9	341[M - H]^−^ 683[2M - H]^−^ 161 [M - H - 180 (hexose)]^−^	365[M + Na]^+^	342	C_15_H_18_O_9_	Caffeoyl hexoside ^49^	nq
3	25.4	609[M - H]^−^ 301[Quercetin - H]^−^	611[M + H]^+^ 325[M + H + K]^2+^ 633[M + Na]^+^	610	C_27_H_30_O_16_	Rutin (Quercetin 3-O-rutinoside) ^st^	nq
4	25.9	463[M - H]^−^ 928[2M - H]^−^ 300[Quercetin - 2H]^−^	465[M + H]^+^ 487[M + Na]^+^ 952[2M + Na]^+^	464	C_21_H_20_O_12_	Quercetin-3-O-hexoside ^48,49,50^	7.0 ± 0.3
5	26.4	463[M - H]^−^ 928[2M - H]^−^ 301[Quercetin - H]^−^	465[M + H]^+^ 487[M + Na]^+^ 952[2M + Na]^+^	464	C_21_H_20_O_12_	Quercetin-3-O-glucoside ^st^	7.4 ± 0.2
6	27.9	593[M - H]^−^ 285[Kaempferol - H]^−^	595[M + H]^+^ 317[M + H+K]^2+^ 617[M + Na]^+^	594	C_27_H_30_O_15_	Kaempferol disaccharide ^48,50^	nq
7	28.1	447[M - H]^−^ 896[2M - H]^−^ 285[Kaempferol - H]^−^	449[M + H]^+^ 471[M + Na]^+^ 920[2M + Na]^+^	448	C_21_H_20_O_11_	Kaempferol hexoside ^48,49,50^	14.7 ± 1.5
8	28.5	609[M - H]^−^	611[M + H]^+^ 325[M + H+K]^2+^ 633[M + Na]^+^	610	C_27_H_30_O_16_	Quercetin disaccharide ^50^	nq
9	28.9	433[M - H]^−^	435[M + H]^+^ 457[M + Na]^+^	434	C_20_H_18_O_11_	Quercetin-3-O-arabinoside ^51^	nq
10	29.3	447[M - H]^−^ 896[2M - H]^−^ 285[M - H - 163 (hexose)]^−^	449[M + H]^+^ 471[M + Na]^+^ 920[2M + Na]^+^	448	C_21_H_20_O_11_	Kaempferol-3-O-glucoside ^49,51^	89.0 ± 0.5
11	31.5	435[M - H]^−^	459[M + Na]^+^ 238[M + H + K]^2+^	436		Unknown	nq
12	31.7	417[M - H]^−^	419[M + H]^+^ 441[M + Na]^+^ 895[2M + Na]^+^	418	C_20_H_18_O_10_	Kaempferol pentoside ^49,50,51^	nq
13	33.1	593[M - H]^−^ 285 [Kaempferol - H]^−^	595[M + H]^+^ 317[M + H + K]^2+^ 617[M + Na]^+^	594	C_27_H_30_O_15_	Kaempferol disaccharide (Kaempferol -O-pentose -O-glucuronic acid) ^48,49,50,51^	19.7 ± 1.5
14	33.9	417[M - H]^−^ 836[2M - H]^−^ 285[M - H - 133 (pentose)]^−^	419[M + H]^+^ 441[M + Na]^+^	418	C_20_H_18_O_10_	Kaempferol pentoside ^49,51^	nq
15	35.9	431 [M - H]^−^ 863 [2M - H]^−^	433[M + H]^+^ 455[M + Na]^+^ 888 [2M + Na]^+^	432	C_21_H_20_O_10_	Kaempferol deoxyhexoside ^50^	17.5 ± 1.5
16	40.8	635[M - H]^−^ 593[Kaempferol disaccharide - H]^−^	637[M + H]^+^ 659[M + Na]^+^ 338 [M + H + K]^+^	636	C_29_H_32_O_16_	Kaempferol acetyldisaccharide ^50^	nq
17	41.7	593 [M - H]^−^	595[M + H]^+^ 617 [M + Na]^+^	594	C_27_H_30_O_15_	Kaempferol disaccharide (Kaempferol-O-hexose-O-deoxyhexose) ^50,53^	nq
18	44.7	285 [M - H]^−^	287 [M + H]^+^	286	C_15_H_10_O_6_	Kaempferol ^st^	nq

Notes nq: not quantified. st: standard compound used for identification. The superscript numbers indicate the previous studies on *Rosa* spp. that report the same ingredient.

**Table 3 pathogens-10-00494-t003:** LC/MS identification of metabolites and their concentration in the extract of rose geranium leaves.

Peak	Rt (min)	Negative Ionization (*m/z)*	Positive Ionization (*m/z*)	M.W.	Molecular Formula	Tentative Identification	C (μg/mL)
1	23.2	595[M - H]^−^ 462[M - H - 132]^−^ 445[M - H -132 -H_2_O]^−^ 300[quercetin - H]^−^	597[M + H]^+^ 619 [M + Na]^+^	596	C_26_H_28_O_16_	Quercetin-3-O-pentosyl hexoside ^54^	15.7 ± 0.2
2	25	609[M - H]^-^ 301[quercetin]^−^ 300[quercetin - H]^−^ 179	611[M + H]^+^ 325[M + H+K]^2+^ 633 [M + Na]^+^	610	C_27_H_30_O_16_	Quercetin-3-O-rhamnoside hexoside ^54,55,56,57^	nq
3	25.5	463[M - H]^−^ 927[2M - H]^−^ 316[Myricetin - 2H]^−^ 317[Myricetin - H]^−^ 287, 179	465[M + H]^+^ 487[M + Na]^+^ 951 [2M + Na]^+^	464	C_21_H_20_O_12_	Myricetin-3-O-rhamnoside ^54^	nq
4	25.8	463[M - H]^−^ 927[2M - H]^−^ 301[quercetin - H]^−^ 300[quercetin - 2H]^−^179	465[M + H]^+^ 487[M + Na]^+^ 951 [2M + Na]^+^	464	C_21_H_20_O_12_	Quercetin-3-O-galactoside ^54^	13.9 ± 1.2
5	26.3	463[M - H]^−^ 927[2M - H]^−^ 301[quercetin - H]^−^ 255, 179	465[M + H]^+^ 487[M + Na]^+^ 951 [2M + Na]^+^	464	C_21_H_20_O_12_	Quercetin-3-O-glucoside ^st^	17.7 ± 0.3
6	27.9	433[M - H]^−^ 867[2M - H]^−^ 300[quercetin - H]^−^ 255	435[M + H]^+^ 457[M + Na]^+^ 891 [2M + Na]^+^	434	C_20_H_18_O_11_	Quercetin 3-O- pentoside ^54^	3.8 ± 0.6
7	28	447[M - H]^−^ 895[2M - H]^−^ 285 [M - H - 163]^−^	449[M + H]^+^ 471[M + Na]^+^ 919 [2M + Na]^+^	448	C_21_H_20_O_11_	Kaempferol 3-O-glucoside ^54,55^	4.7 ± 0.0
8	29.3	447[M-H]^−^ 895 [2M-H]^−^	449[M + H]^+^ 471[M + Na]^+^ 919[2M + Na]^+^	448	C_21_H_20_O_11_	Kaempferol 3-O-galactoside ^52^	nq
9	30.6	417 [M - H]^−^	419[M + H]^+^ 441 [M + Na]^+^	418	C_20_H_18_O_10_	Kaempferol 3-O- pentoside ^54^	nq
10	24.9	507[M + Formic Acid - H]^−^	485 [M + Na]^+^	462	C_21_H_18_O_12_	Scutelarein-7-O-β-glucuronide ^58^	nq
11	40.1	313 [M - H]^−^	315[M + H]^+^ 651 [2M + Na]^+^	314	C_17_H_14_O6	Cirsimaritin ^58^	nq

Notes nq: not quantified, st: standard compound used for identification. The superscript numbers indicate the previous studies on leaves of *Pelargonium* spp. that mention the same ingredient.

**Table 4 pathogens-10-00494-t004:** Results from essential oil and plant extracts toxicity experiments.

EOs	NEG CONTROL	*Pelargonium graveolens* (5%)	*Rosa damascena* (5%)	POS CONTROL
ΤA98	Nontoxic	Nontoxic	Nontoxic	Toxic
ΤA100	Nontoxic	Nontoxic	Nontoxic	Toxic
**Extract**	**NEG CONTROL**	***Pelargonium graveolens* (90%)**	***Rosa damascena* (90%)**	**POS CONTROL**
ΤA98	Nontoxic	Nontoxic	Nontoxic	Toxic
ΤA100	Nontoxic	Nontoxic	Nontoxic	Toxic

**Table 5 pathogens-10-00494-t005:** Percentage of essential oils and extracts inhibitory activity against bacteria at different values of final content in growth medium (from 5 to 100% for essential oils and 25 to 90% for extracts).

Essential Oils	*E. coli*	*S. aureus*	*Salmonella* spp.
*Pelargonium graveolens* (100%)	85%	55%	51%
*Pelargonium graveolens* (50%)	29%	34%	31%
*Pelargonium graveolens* (5%)	-	6%	13%
*Rosa damascena* (100%)	65%	42%	86%
*Rosa damascena* (50%)	36%	40%	73%
*Rosa damascena* (5%)	2%	28%	22%
**Extracts**			
*Pelargonium graveolens* (90%)	68%	40%	42%
*Pelargonium graveolens* (50%)	59%	28%	33%
*Pelargonium graveolens* (25%)	46%	14%	17%
*Rosa damascena* (90%)	74%	68%	86%
*Rosa damascena* (50%)	57%	43%	65%
*Rosa damascena* (25%)	35%	37%	45%

Note: “-” indicates that the essential oil had no inhibitory activity on the tested strain at this concentration. 0–25%, no or little inhibition; 26–50%, average inhibition; 51–75%, strong inhibition. According to the CLSI breaking points for a given inhibitor concentration: % inhibition = 100 * [1 − (x − min)/(max − min)], Humphries et al., 2019 [39].

**Table 6 pathogens-10-00494-t006:** Effects of cytotoxicity of essential oils.

Essential Oil	Effect on Cell Line A549	Extract	Effect on Cell Line A549
*Pelargonium graveolens* 100%	Cytotoxic	*Pelargonium graveolens* 100%	Non-cytotoxic
*Pelargonium graveolens* 5%	Noncytotoxic	*Pelargonium graveolens* 90%	Non-cytotoxic
*Rosa damascena* 100%	Cytotoxic	*Rosa damascena* 100%	Non-cytotoxic
*Rosa damascena* 5%	Noncytotoxic	*Rosa damascena* 90%	Non-cytotoxic

**Table 7 pathogens-10-00494-t007:** Effects of essential oils and extracts on Adenovirus.

Essential Oil 5%	AdV10^9^ PFU/mL	AdV 10^8^ PFU/mL	AdV10^7^ PFU/mL	AdV10^6^ PFU/mL	AdV10^5^ PFU/mL	Adv10^4^ PFU/mL
*Pelargonium graveolens*	+	+	+	+	+	+
*Rosa damascena*	-	-	+	+	+	+
**Extract**						
*Pelargonium graveolens* 90%	-	+	+	+	+	+
*Pelargonium graveolens* 50%	-	+	+	+	+	+
*Pelargonium graveolens* 25%	-	-	+	+	+	+
*Rosa damascena* 90%	+	+	+	+	+	+
*Rosa damascena* 50%	+	+	+	+	+	+
*Rosa damascena* 25%	+	+	+	+	+	+

Note: +: Effect against Adenoviruses, -: No effect against Adenoviruses.

## Data Availability

Data is contained within the article.

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
