# Peer review of "Evaluation of Essential Oils and Extracts of Rose Geranium and Rose Petals as Natural Preservatives in Terms of Toxicity, Antimicrobial, and Antiviral Activity"

_pathogens, 2021, doi:10.3390/pathogens10040494_

Round 1

Reviewer 1 Report

Manuscript entitled “Evaluation of Essential Oils and Extracts of rose geranium and rose petals as natural preservatives in terms of toxicity, antimicrobial and antiviral activity” describes results that contribute to understand potential activity of Essential Oils and Extracts of rose geranium and rose petals as novel antimicrobial strategies.

Although this is a very interesting study, there are important points that need addressing.

  1. Photomicrographs showing control and infected cells, exposed or not to all compounds, may help to further elucidate the antiviral effects of EOs.
  2. Have you established a limit of quantification (LOQ) for the main components of both rose geranium and rose petals? Please, specify in manuscript
  3. Have you evaluated pesticides or other environmental contaminants in your Essential Oils and Extracts? Please, specify in manuscript

Other points are summarized below.   

Introduction

Line 50-52, rewrite this sentence.

Materials and Methods

Line 83, specify the solvent.

Line 171, change 24h in 24 h, here and throughout the manuscript.

Line 311, added the subject in the sentence.

Discussion

Line 366-369, specify your climate conditions

Line 369, change “in conclusion”

Line 406, specify the receptors.

Line 409, specify some microorganisms.

Line 422-425, specify some mechanism of action.

Please, change ml, µl… in mL, µL…throughout the manuscript.

Author Response

Manuscript entitled “Evaluation of Essential Oils and Extracts of rose geranium and rose petals as natural preservatives in terms of toxicity, antimicrobial and antiviral activity” describes results that contribute to understand potential activity of Essential Oils and Extracts of rose geranium and rose petals as novel antimicrobial strategies.

Although this is a very interesting study, there are important points that need addressing.

  1. Photomicrographs showing control and infected cells, exposed or not to all compounds, may help to further elucidate the antiviral effects of EOs.

…Unfortunately we had not taken any photos of these

  1. Have you established a limit of quantification (LOQ) for the main components of both rose geranium and rose petals? Please, specify in manuscript

We have expressed the concentration of all ingredients as rutin equivalents and thus established LLOQ and LLOD limits for rutin standard. The relevant details are now described in lines 162-165.

Text: “The Lower Limit of Quantitation (LLOQ) was 3.125 μg/mL since it is the lowest concentration of rutin giving an acceptable accuracy (relative error <20%) and the Lower Limit of Detection (LLOD) was 0.875 μg/mL calculated as a signal to noise ratio of 3.”

  1. Have you evaluated pesticides or other environmental contaminants in your Essential Oils and Extracts? Please, specify in manuscript 

Although we did not perform specialized analytical methods for pesticides or other environmental contaminants, the evaluation of the chromatograms and of the mass spectra of each peak did not show the presence of any such peaks. This is now discussed in lines 268-269.

Text: “The evaluation of the mass spectra of each chromatographic peak did not allude to the presence of any pesticide or other contaminants in the samples.”

Other points are summarized below.   

Introduction

Line 50-52, rewrite this sentence.

It is now rewritten in lines 51-54.

Original: “The main compounds in essential oils are oxygenated monoterpenes (64,3-74,2%), albeit its composition greatly depends on the genotype, the environmental conditions, the agricultural practices of cultivation and the time of collection [13].”

Revised: “ The main volatile compounds are oxygenated monoterpenes (64,3-74,2%), but the essential oil composition greatly depends on the genotype, the environmental conditions, the agricultural practices of cultivation and the time of collection [13].”

Materials and Methods

Line 83, specify the solvent.

It is now specified in lines 84 and 86-87.

“Fresh rose petals were dried for 1 day and 2 kg of plant material were extracted for 35 days, at a temperature of 22-27°C, in 40 L of water containing 15% v/v ethanol. Fresh geranium leaves were dried for 8 hours and then 0.2 kg of plant material were extracted for 30 days, at a temperature of 22-27°C, in 40 L of 38% v/v aqueous ethanol.”

Line 171, change 24h in 24 h, here and throughout the manuscript.

It is done throughout the text, accordingly.

Line 311, added the subject in the sentence.

It is done throughout the text, accordingly.

Discussion

Line 366-369, specify your climate conditions

A description of the climate conditions of the collection area is now provided in lines 433-435, as follows:

Text: “Accordingly, in our study, rose petals were harvested in late spring from a mountainous location (360 m altitude) at a Northern Aegean island which enjoys mild Mediterranean climate; in late spring the mean temperature does not exceed 20°C.”

Line 369, change “in conclusion”

It is now changed with “Therefore” (line 435)

Line 406, specify the receptors.

Corrected in the text

Line 409, specify some microorganisms.

Added in the text, “such as Candida or Staphylococcus”

Line 422-425, specify some mechanism of action.

A relevant comprehensive review has now been added (reference 70) and a general description of the mechanisms of action is now provided in lines 496-501, as follows:

“The antiviral properties of quercetin and kaempferol derivatives have been reported in numerous publications [70], e.g. against influenza viruses [71, 72], against coronavirus and dengue viruses [73, 74], and in particular of quercetin against adenoviruses 1 and 3 [75]; the reports indicate that those flavonols block viral entry to the host cell via specific interactions with viral attachment factors and/or membrane fusion proteins, suppress signaling pathways that are essential for virus gene expression, inhibit remodelling enzymes and channels which regulate viral movement (e.g. neuraminidase), and inhibit transcription of the viral genome and viral protein synthesis [70].”

Please, change ml, µl… in mL, µL…throughout the manuscript.

 Ιt has been corrected accordingly.

Reviewer 2 Report

The article ‘Evaluation of Essential Oils and Extracts of rose geranium and rose petals as natural preservatives in terms of toxicity, antimicrobial and antiviral activity’ by Andritsopoulou et al is a detailed and elaborate study on the characterization and the properties of essential oil and extracts from two plants that are commonly grown in Greece.

Minor corrections need to be made:

  1. Figure 2. Antifungal activity through time of Extracts at concentrations of 90 (a), 50 (b) and 25 (c) % ------------------------should be expressed as:

Antifungal activity through time of extracts at concentrations of 90% (a), 50% (b) and 25% (c)

  1. All bacteria and fungus nomenclature should be written in italics. e.g Escherichia coli
  2. Table 5. Percentage of essential oils and extracts growth inhibitory activity against bacteria at

different values of final content in growth medium (from 5 to 100% for essential oils and 25 to 90% 263

for extracts). --------------------------------------------------------- remove ‘growth’

Author Response

The article ‘Evaluation of Essential Oils and Extracts of rose geranium and rose petals as natural preservatives in terms of toxicity, antimicrobial and antiviral activity’ by Andritsopoulou et al is a detailed and elaborate study on the characterization and the properties of essential oil and extracts from two plants that are commonly grown in Greece.

Minor corrections need to be made:

  1. Figure 2. Antifungal activity through time of Extracts at concentrations of 90 (a), 50 (b) and 25 (c) % ------------------------should be expressed as:

Antifungal activity through time of extracts at concentrations of 90% (a), 50% (b) and 25% (c)

It has now been changed as suggested by the reviewer in lines 356.

  1. All bacteria and fungus nomenclature should be written in italics. e.g Escherichia coli

Thank you for bringing to our attention this detail. We have corrected it throughout the text.

  1. Table 5. Percentage of essential oils and extracts growth inhibitory activity against bacteria at different values of final content in growth medium (from 5 to 100% for essential oils and 25 to 90% for extracts). --------------------------------------------------------- remove ‘growth’

It has now been deleted in line 312…
